# Preliminary Assessment of Chemical Elements in Sediments and Larvae of Gomphidae (Odonata) from the Blyde River of the Olifants River System, South Africa

**DOI:** 10.3390/ijerph17218135

**Published:** 2020-11-04

**Authors:** Abraham Addo-Bediako, Karabo Malakane

**Affiliations:** Department of Biodiversity, University of Limpopo, Private Bag X1106, Sovenga 0727, South Africa; karabo.malakane@ul.ac.za

**Keywords:** bioaccumulation, Gomphidae, heavy metals, naiads, metalloids, pollution, sediments

## Abstract

Benthic macroinvertebrates and sediments can act as good indicators of environmental quality. The aim of this study was to assess the accumulation of chemical elements in the Gomphidae (Odonata) collected in the Blyde River. Seven sites were sampled for river sediments assessment and five sites for larvae (naiads) of Gomphidae bioaccumulation analysis. The tissue samples were analysed using inductively coupled plasma optical emission spectrometry (ICP-OES). The results showed high levels of all of the tested elements except Cd in the sediment. The mean concentrations of As, Cu and Cr exceeded the standard guideline values, whereas Pb and Zn were below the standard guideline values. In the insect body tissue, the concentrations of most elements were higher than in the sediments. The elements with the highest concentrations were Mn, Zn, Cu, and As. The bioaccumulation factor (BF) showed a tendency for bioaccumulation for almost all of the selected elements in the insect. The BF value was high for Cu, Mn, Sb, and Zn (BF > 1). The high concentrations of elements in the insect body tissue may pose a risk to fish that consume them, and subsequently to humans when fish from the river are consumed. It is therefore important to monitor the river to reduce pollution to prevent health risks in humans, especially in communities that rely on the river for water and food.

## 1. Introduction

Globally, rivers and streams are threatened by anthropogenic pollution, such as toxic elements, due to intensive land-use and inadequate environmental management practices [1,2,3]. Though most elements occur naturally in the biogeochemical cycle, many are released into inland waters as industrial, mining, agricultural, and domestic effluents, and may be harmful to aquatic systems [4]. River sediments serve as a habitat for various benthic macroinvertebrates and can serve as a sink for elements such as heavy metals. The burrowing activity of some benthic organisms leads to their chronic exposure to sediments contaminated with chemical elements [5].

Some elements are essential micronutrients for living organisms, while some (e.g., Cd, Cr and Pb) are toxic to living organisms, even at low concentrations. The toxicity of elements in aquatic ecosystems is complex and dependent on their bioavailability. Due to their prevalence and toxicity, heavy metal contamination in aquatic ecosystems poses a serious environmental threat [6,7,8]. This may lead to a decline in freshwater ecosystem functioning and biodiversity [9]. The available elements in the environment (sediment and water) can be assimilated into living tissues through direct uptake and the food chain, and if accumulated at unacceptable concentrations can affect the aquatic biota [10]. When the contaminants are incorporated into the food chain, it poses a toxicity risk to the organisms that consume them: fish, fish-eating birds, mammals and humans [11].

Many benthic organisms represent a link for the transfer of elements from the sediments to upper trophic levels. Macroinvertebrates play a major ecological role in conveying energy from lower trophic levels upwards. They serve as food for many predatory organisms in the water including fish, which are a vital food for many rural communities, especially low-income groups [12]. Humans who regularly consume contaminated fish are at risk to genotoxic, carcinogenic, and non-carcinogenic health impairment from long-term exposure to toxic contaminants [13,14]. Thus, it has become increasingly important to assess the levels of chemical elements in the body tissues of aquatic organisms as an indicator of metal and metalloid pollution in aquatic systems and to determine whether the food (e.g., fish) from impacted river systems are suitable for human consumption [15].

The Blyde River is one of the main tributaries of the Olifants River System. The river serves as a source of drinking water and food (fish) to the rural communities living in the catchment. The larvae (naiads) of dragonflies (Gomphidae, order Odonata) were selected for the study. They are good ecological indicators and reflect the quality of aquatic systems [16,17]. The larvae are important predators in aquatic ecosystems and prey on benthic and planktonic invertebrates [18] and also serve as food for many fish species. The aim of the study was to assess the concentration of chemical elements (bioaccumulation) in the larvae of Gomphidae and to predict the potential risk of transfer of toxic elements to fish species.

## 2. Materials and Methods

### 2.1. Study Area

The Blyde River rises on the western slopes of the north-south trending Drakensberg Mountains and flows northwards towards the escarpment edge where it is dammed. From the dam, the Blyde River cascades down a steep series of rapids to its lower reaches, where the river again flows northwards to join the Olifants River at the town of Hoedspruit in Limpopo Province [19]. The Blyde River sub-catchment is approximately 2000 km^2^ in size. Geologically, the northern part of the sub-catchment is made up of crystalline gneissic and granitic rocks of the Basement Complex, underlying the catchment [19]. The sub-catchment lies partly on the escarpment and, as a result, experiences considerably higher rainfall than the other sub-catchments in the Olifants River Basin, with mean annual precipitation sometimes exceeding 1000 mm [19]. During the last decade, there has been an increase in human activities in the area, especially agriculture, which are likely to cause environmental pollution in the freshwater systems.

The river is subjected to various sources of anthropogenic pollution, including domestic waste (S1 and S2), agricultural runoff (S3 and S5), and industrial waste (Site 4), while S6 and S7 are nature reserves (Table 1). The sampling sites were spread along the Blyde River until near the confluence with the Olifants River. The study sites ranged between 24°30′59.46” S 30°47′56.14” E and 24°15′30.38” S 30°50′13.22” E (Figure 1).

### 2.2. Sampling and Analysis

Sediment samples were collected at seven sites along the Blyde River during the months of February, April, July and October, in 2018. The samples were collected in acid pre-treated polyethylene bottles. The sediment was frozen prior to chemical analysis. Gomphidae larvae were sampled using a 30 by 30 cm SASS net with a 500 µm mesh size [20]. The samples collected at S3 and S7 were not sufficient for chemical analysis. Sediments and macroinvertebrate samples were then analysed for elements at an accredited (ISO 17025) chemical laboratory (WATERLAB (PTY) LTD, Pretoria, South Africa). The samples were put in acid-washed polypropylene pre-weighed vials and dried at 60 °C for 24 h, and a mixture of HNO_3_ and HCl was added. Subsequently, the samples were digested in an oven [21]. The digested samples were cooled at room temperature, filtered using filter papers, and collected in beakers. The following metals and metalloids were then analysed in batches with blanks using inductively coupled plasma–optical emission spectrometry (ICP-OES; Perkin Elmer, Optima 2100 DV, Pretoria, South Africa): Arsenic (As), Antimony (Sb), Cadmium (Cd), Chromium (Cr), Copper (Cu), Lead (Pb), Manganese (Mn), Nickel (Ni) and Zinc (Zn). The analytical accuracy was determined using certified standards (De Bruyn Spectroscopic Solutions 500 MUL20 - 50 STD2) and recoveries were within 10% of certified values. The detection limits were: As—0.001 mg/kg, Cd—0.0001 mg/kg, Cr—0.001 mg/kg, Cu—0.001 mg/kg, Mn—0.0025 mg/kg, Ni—0.001 mg/kg, Pb—0.001 mg/kg, Sb—0. 001 mg/kg, and Zn—0.001mg/kg.

### 2.3. Statistical Analysis

The mean and standard deviation of four samples at each site from the respective concentrations of the elements in the sediments were calculated. Analysis of variance (ANOVA) was performed using SPSS to determine whether there were significant differences among the different sites for the concentrations of the elements. Pearson’s correlation matrix was used to identify the relationship between the metals. The ability of benthic macroinvertebrates to accumulate chemical elements was quantified through the bioaccumulation factors (BF) according to Klavinš et al. [22]

The bioaccumulation factor is calculated using the following formula:BF = C_org_/C_sediment_
where C_org_ is the element mass fraction in the organism (mg kg^−1^ dry weight) and C_sediment_ is the element concentration of the sediment (mg kg^−1^ dry weight).

## 3. Results and Discussion

The mean concentrations of the elements in the sediment samples at the different sites are shown in Table 2. The concentrations of As, Cu and Sb varied significantly among the different sites (*p* < 0.05). The variations in the concentrations of the elements among sites could be due to the type of effluents washed into the river from the catchment. The highest concentrations of As, Cu, Sb, and Zn were recorded at S3. The highest concentrations of Cr, Mn and Ni were recorded at S5, and the highest concentrations of Cd at S6.

The high concentrations of most of the chemical elements may be due to direct or indirect land surface runoff of agricultural fields at S3 and the release of urban sewage and industrial effluents at S5 [23,24]. Furthermore, the grain-size distribution of the sediments at different sites could have also contributed to the type and concentrations of the elements. The proportion of finer particles at S5 was higher than that of coarse grains and may have contributed to the high concentration of chemical elements. Thus, as the grain size decreases, the metal content increases [25,26]. The mean concentration of As was greater than the CCME [27] guideline value of 13 mg kg^−1^, dw at all the sites. The high As concentration at S3 might have been coming from pesticides and fertilizers used in agricultural fields [28,29]. The mean concentrations of Cr exceeded the guideline value of 37.3 mg kg^−1^, dw at all the sites. Chromium and its salts are used in pigments and paints, in fungicides, and in chrome alloy and chromium metal production [30]. In this study, the main source of Cr in the sediment was mainly from agricultural activities. The concentration of Cu exceeded the guideline value of 37.3 mg kg^−1^, dw at all the sites except S7. The high concentration of Cu in the study sites could be attributed to agricultural activities (pesticides, herbicides and fungicides) and to municipal wastewater and discharges from the catchment.

The correlation matrix showed a very strong relationship between Cr and Ni (0.868), Cu and Zn (0.897), and Pb and Zn (0.766), at a significance level of 0.01. There was a strong relationship between As and Cr (0.635), Cd and Cu (0.760), Cr and Mn (0.679), Ni and Mn (0.750), and Cd and Zn (0.727) at a significance level of 0.05 (Table 3). These results indicated that these elements originated from similar pollution sources. The absence of a correlation among some of the elements suggests that they are not controlled by a single factor [31]. The high concentrations of these elements in the sediments may pose an ecological risk to the aquatic biota, especially bottom-dwelling organisms. The concentration of Cd was very low in the river. The relatively low levels of the elements at the downstream sites (S6 and S7) is attributed to the nature conservation practices at these two sites. This is an indication that the conservation practice is having a positive impact on the downstream of the river.

The tissue of Gomphidae (Odonata) was analysed for these chemical elements; As, Cd, Cr, Cu, Mn, Ni, Pb, Sb and Zn. Aquatic insects can accumulate pollutants such as heavy metals from stream sediments and from food [32,33]. There were significant differences in the concentrations of Mn, Ni, Pb and Zn recorded in the body tissues of the insect larvae (*p* < 0.05). The concentrations of the elements in the body tissues varied among the sites, with the highest concentrations of all the elements with the exception of Mn and Ni were at S1. The concentrations of most of the elements in the aquatic insect were about five to 10 times those of the sediments. The larvae bioaccumulated lower concentrations at the downstream site, S6 (Table 4). The highest bioaccumulation of elements was at S1, instead of S3 or S5, which had the highest concentrations of most of the elements in the sediments and could partly be due to the local bioavailability of these elements.

Most of the elements detected in high concentrations in the sediments and in the insect larvae are widely used in several fertilizers as a source of micronutrients. The larvae of Odonata are known to tolerate heavy metals [34]. The concentrations of Cd, Cu, Zn and Mn were found in higher concentrations (>50%) in the insect tissue than in the sediment. Meanwhile, the concentrations of As, Cr, Ni, Pb and Sb were higher in the sediments than in the tissue of the larvae (Figure 2). The elements in high concentrations in the sediments, such as Mn, Cu and Zn, were highly bioaccumulated in the insects. In this study, the transfer of Cr, Ni and Sb into the body tissue of the insect larvae was relatively less efficient, whereas Cu, Mn and Zn showed relatively high transfer efficiency. In aquatic insects, the concentrations of Cd, Ni, Cr, As, Pb, Cu, Ti, Zn and Mn change with size, life cycle stages, and different bioaccumulation patterns [35]. For example, Caddisflies have been found to accumulate Pb, regulate Zn and Cu, while Stoneflies accumulate Pb and regulate Zn [36].

The bioaccumulation factor (BF) of the elements in the insect larvae of Gomphidae from the Blyde River are shown in Figure 3. The BF value was >1 for Cu, Mn, Sb and Zn, thus these elements may be transferred to fish, and then to humans who consume fish from the river. The BF was high at the upstream sites, S1 and S2, indicating a high bioavailability of the elements for the insect larvae, whereas the lowest BF was at S6 (downstream site), with relatively low concentrations of the elements in the sediments. The results show that the larvae of Gomphidae accumulate chemical elements from the environment and they can be used to detect metal and metalloid pollution in aquatic environments [37].

## 4. Conclusions

The metal and metalloid analysis of the river sediments showed variations in their concentrations among the sites. The effects of these elements may have consequences not only on aquatic insects, but also on higher trophic levels, such as fish and humans. In the insect body tissue, the concentrations of most of the chemical elements were higher than in the sediments, meaning that the insects accumulated the elements from the sediments. The study suggests that the concentrations of many of the elements studied are too high in the sediment and the larval tissue; it is therefore necessary to monitor and control chemical pollution in the river. Further study is required to assess the level of accumulation in the different functional groups of macroinvertebrates and to determine the transfer of toxic elements through the food chain.

## Figures and Tables

**Figure 1 ijerph-17-08135-f001:**
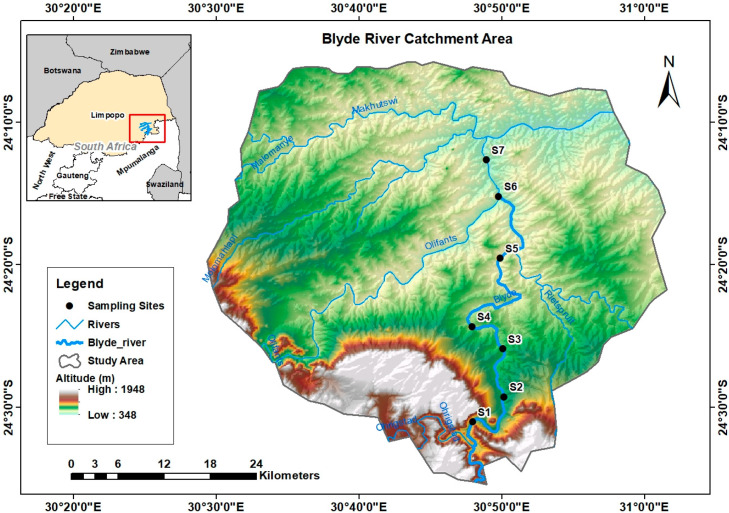
Map of the study area, showing the locations of the seven sampling sites of the Blyde River.

**Figure 2 ijerph-17-08135-f002:**
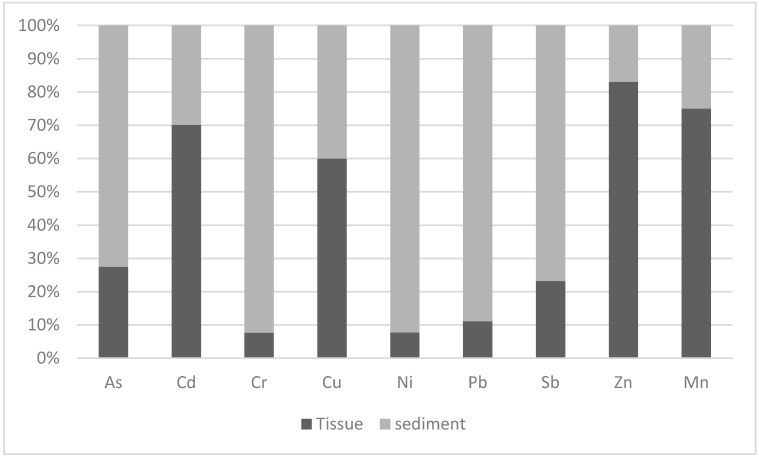
Composition of chemical elements in the sediments and the tissue of Gomphidae larvae.

**Figure 3 ijerph-17-08135-f003:**
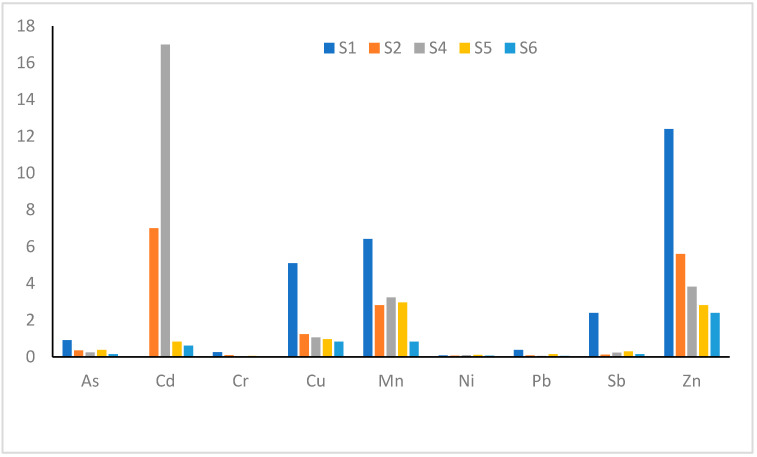
Bioaccumulation factor (BF) for larvae of Gomphidae samples from the Blyde River (ratio of concentrations of chemical elements in the larval tissue and in the sediment).

**Table 1 ijerph-17-08135-t001:** Location, description of activities, vegetation cover and substrate type (%).

Site	Activity	Vegetation Cover	Cobbles	Sand	Silt	Mud
S1	Domestic	70% (mainly shrubs and trees)	50	20	20	10
S2	Domestic/agriculture	60% (mainly shrubs, grass, and a few trees)	30	30	20	20
S3	Agriculture (mainly mangoes and citrus)	90% (mainly trees and shrubs)	40	30	20	10
S4	Industries (mainly local furniture manufacturing, automotive services and fruit processing factories)	20% (mainly shrubs and grass)	30	20	30	20
S5	Agriculture (mainly mangoes and citrus)	70% (mainly trees and shrubs)	20	20	30	30
S6	Nature reserve (little human activity)	80% (mainly trees, shrubs and grass)	20	20	30	30
S7	Nature reserve (little human activity)	50% (mainly shrubs and grass, and a few trees)	30	30	20	20

**Table 2 ijerph-17-08135-t002:** Concentrations (mg kg^−1^) of chemical elements at different sites in the Blyde River sediment samples.

Element	S1		S2		S3		S4		S5		S6		S7		SQG
	AVE	± SD	AVE	±SD	AVE	±SD	AVE	±SD	AVE	±SD	AVE	±SD	AVE	±SD	
As	29.04	19.6	57.2	59.2	107.57	49.3	51.03	40.5	44.88	46.6	50.79	46.0	6.23	3.6	5.9
Cd	ND	-	0.04	0.05	0.09	0.1	0.01	0.02	0.11	0.18	0.41	0.7	ND	-	0.6
Cr	56.33	15.5	48.9	16.8	98.24	42.5	80.44	50.3	108.0	73.8	41.5	12.8	76.1	449	37.3
Cu	36.74	20.4	82.0	90.2	274.34	148.3	73.99	50.8	63.46	52.8	63.62	62.1	15.23	8.9	35.7
Mn	494.6	69.1	748.7	530.4	1175	490.5	949.8	635	1298.8	776	685.31	263	984.3	404	-
Ni	137.4	118	126.9	111.2	166.4	104.8	115.1	101	281.69	329	109.9	104	288.1	301	-
Pb	4.94	0.57	7.23	1.68	16.13	4.1	7.18	1.83	7.36	1.1	7.49	2.1	6.57	0.75	35
Sb	1.48	1.1	8.19	7.71	24.74	6.7	6.3	5.6	7.24	7.0	7.11	5.4	0.4	0.69	-
Zn	29.19	24.2	30.1	22.19	75.68	62.2	48.26	25.0	38.58	23.6	42.83	38.7	45.58	40.9	123

AVE: Average; SD: standard deviation; ND—not detected. SQG: Sediment quality guideline (CCME).

**Table 3 ijerph-17-08135-t003:** The correlation coefficients between chemical elements of the sediments in the Blyde River.

Element	Sb	As	Cd	Cr	Cu	Pb	Mn	Ni	Zn
Sb	1	0.217	0.103	0.099	−0.271	−0.389	0.574	0.452	−0.289
As		1	0.111	0.635	0.259	−0.342	0.597	0.386	−0.161
Cd			1	−0.452	0.760	0.368	−0.110	−0.552	0.727
Cr				1	−0.086	−0.500	0.679	0.868	−0.392
Cu					1	0.586	−0.202	−0.386	0.897
Pb						1	−0.531	−0.628	0.766
Mn							1	0.750	−0.467
Ni								1	−0.556
Zn									1

**Table 4 ijerph-17-08135-t004:** Concentration of chemical elements (Mean ± S.E) in the tissue of Gomphidae (Odonata) larvae at different sites of the Blyde River (S.E: standard deviation).

Element	S1	S2	S4	S5	S6
As	32.26 ± 0.0	19.81 ± 2.4	12.32 ± 0.0	16.59 ± 3.5	7.3 ± 3.3
Cd	0.56 ± 0.0	0.28 ± 0.03	0.17 ± 0.0	0.09 ± 0.03	0.25 ± 0.25
Cr	13.81 ± 0.0	4.59 ± 1.78	2.05 ± 0.0	5.55 ± 2.2	1.82 ± 0.53
Cu	187.13 ± 0.0	101.07 ± 28.1	78.18 ± 0.0	61.1 ± 29.2	52.9 ± 26.6
Mn	3173 ± 0.0	2106 ± 395	3068 ±0.0	3637 ± 1038	563.2 ± 33.6
Ni	11.13 ± 0.0	8.17 ± 2.79	9.99 ± 0.0	29.47 ± 10.3	6.23 ± 5.7
Pb	1.9 ± 0.0	0.55 ± 0.1	0.33 ± 0.0	1.11 ± 0.6	0.38 ± 0.05
Sb	3.54 ± 0.0	0.97 ± 0.05	1.46 ± 0.07	2.18 ± 0.95	1.03 ± 0.17
Zn	362.2 ± 0.0	168.2 ± 3.9	183.8 ± 0.0	108.7 ± 57.6	102.3 ± 41.2

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
