# Peer review of "Preliminary Assessment of Chemical Elements in Sediments and Larvae of Gomphidae (Odonata) from the Blyde River of the Olifants River System, South Africa"

_ijerph, 2020, doi:10.3390/ijerph17218135_

Round 1

Reviewer 1 Report

The manuscript “Preliminary assessment of heavy metals in sediments 2 and larvae of Gomphidae (Odonata) from the Blyde 3 River of the Olifants River System, South Africa” deals with an important environmental topic. Although the number of samples is limited, the research seems to have been conducted in an honest way.

It appears that the manuscript, which is anomaly short, was prepared in very hasty way. The end of the abstract and the authors contribution part show good signs of it.

In terms of science, there are a few issues that have to be carefully addressed before publishing in an international journal. Namely.

Materials and Methods

- The description of the study area is too limited. And this compromises the interpretations made on the Results ad Discussion, some emphasized in the Conclusions. Necessary to add paragraphs on the forms of human occupation and their distribution.

- For each result it is presented a standard deviation, but it is not clear how many samples were analysed or measurements were conducted for each sampling site.

Results and Discussion

- Sediment geochemistry strongly depends on grain-size, with most elements being found in low concentrations in sands, that tend to be enriched in silica. Geochemistry can also be linked with the geology of the source area, which can be revealed by mineralogical data, but readers receive no information about the type of sediments. If no mineral data is available, at least one paragraph/figure describing the sediment texture, namely the proportions of main size-classes (gravel, sand, silt, mud) should be added.

- The interpretation presented in the 1st of Results and Discussion are not well supported. To obtain such conclusion it is necessary to know better the forms of land occupation. In addition, sediment geochemistry can be determined by the geology of the source area and other natural factors, as shown in the article of Dinis et al. (published in the Special Issue that received the present submission; maybe you should cite this work).

- Are there any correlations between the proportions of the investigated elements in sediment and tissues?

Figures

They should be improved. A couple more could be added, as mentioned before.

- Figure 1: Not an adequate figure for an international publication. It could be OK in the 80s or 90s, but is no any longer when so many information is available on natural contexts Needs more information on the environmental characterization and geographic location, as indicated in the annotated manuscript.

Figure 2: Should be reformulated (see suggestion in the attachment)

An annotated manuscript with additional comments and suggestions was attached for authors.

Taking into consideration the above-mentioned points, I recommend major revision before acceptance. I hope that this assessment will help to improve the article.

Author Response

Figure 2 is important results for this study, but any comment is not
found in the manuscript. The authors are encouraged to describe how to
determine the results shown in Figure 2.

Comments on Figure 2 have been addressed in the results and discussion section Paragraph 4.

Use a figure to show the results in Table 3.

A figure (Figure 3) is used for the results of the Bioaccumulation Factor, in Table 3 of the previous version.

Add a summary for the digestion method of Bervoets and Blust.

A summary of the digestion method of Bervoets and Blust has been included in the materials and methods section (2.2).

Add detection limits for each concentration determined from ICP-OES.
Actually, ICP is not recommended for the analysis of Hg. This is why Hg
is not detected in the sediment.

Detection limits of metal concentrations have been added (Section 2.2). Hg is removed from the results.

Reviewer 2 Report

It is well written. However, the following revisions are recommended for publication:

Figure 2 is important results for this study, but any comment is not found in the manuscript. The authors are encouraged to describe how to determine the results shown in Figure 2.

Use a figure to show the results in Table 3.

Add a summary for the digestion method of Bervoets and Blust.

Add detection limits for each concentration determined from ICP-OES. Actually, ICP is not recommended for the analysis of Hg. This is why Hg is not detected in the sediment.   

Author Response

Reviewer 2

Materials and Methods

- The description of the study area is too limited. And this compromises
the interpretations made on the Results ad Discussion, some emphasized
in the Conclusions. Necessary to add paragraphs on the forms of human
occupation and their distribution.

More information is added to the study area (2.1).

- For each result it is presented a standard deviation, but it is not
clear how many samples were analysed or measurements were conducted for
each sampling site.

Four samples were taken during the months of February, April, July and October, 2018

Results and Discussion

- Sediment geochemistry strongly depends on grain-size, with most
elements being found in low concentrations in sands, that tend to be
enriched in silica. Geochemistry can also be linked with the geology of
the source area, which can be revealed by mineralogical data, but
readers receive no information about the type of sediments. If no
mineral data is available, at least one paragraph/figure describing the
sediment texture, namely the proportions of main size-classes (gravel,
sand, silt, mud) should be added.

A paragraph describing the sediment texture at the different sites is included in the Materials and methods (2.1).

- The interpretation presented in the 1st of Results and Discussion are
not well supported. To obtain such conclusion it is necessary to know
better the forms of land occupation. In addition, sediment geochemistry
can be determined by the geology of the source area and other natural
factors, as shown in the article of Dinis et al. (published in the
Special Issue that received the present submission; maybe you should
cite this work).

The information about the human activities occurring along the river is given in the materials and methods (2.1) to support the description in the first paragraph of the results and discussion. The description of the sediments and human activities in each of the sampling site are given.

- Are there any correlations between the proportions of the investigated
elements in sediment and tissues?

Correlation of metals in the sediments have been added (Table 2).

Figures

They should be improved. A couple more could be added, as mentioned before.

- Figure 1: Not an adequate figure for an international publication. It
could be OK in the 80s or 90s, but is no any longer when so many
information is available on natural contexts Needs more information on
the environmental characterization and geographic location, as indicated
in the annotated manuscript.

Figure 1 has been improved. Another figure (Figure 3) on Bioaccumulation factor (BF) is added.

Figure 2: Should be reformulated (see suggestion in the attachment)

Figure 2 is redrawn and more information is added in the text.

Round 2

Reviewer 1 Report

The manuscript was re-submitted after some revision, but as far as I see it, most of my comments were ignored. I am attaching an annotated document where I insist in some point already mentioned and raise a few more. I can give up of some comments/suggestions mentioned in the previous reviewing round, but I recommend that authors give a new look to them.

Sorry if I am not as positive as you may expected.

Author Response

Dear Sir/Madam,

I have addressed all the comments for version 1 and version 2. I could not get access to the track changes you made in the first version.

Thank you for your valuable comments and suggestions.

Kind regards,

Abraham

Reviewer 2 Report

The manuscript has been revised. 

Author Response

The comments have been addressed.